# Tunable Gain SnS_2_/InSe Van der Waals Heterostructure Photodetector

**DOI:** 10.3390/mi13122068

**Published:** 2022-11-25

**Authors:** Seyedali Hosseini, Azam Iraji zad, Seyed Mohammad Mahdavi, Ali Esfandiar

**Affiliations:** 1Center for Nanoscience and Nanotechnology, Institute for Convergent Science and Technology, Sharif University of Technology, Tehran 11155-9161, Iran; 2Department of Physics, Sharif University of Technology, Tehran 11155-9161, Iran

**Keywords:** photodetector, two-dimensional material, Van der Waals heterostructure

## Abstract

Due to the favorable properties of two-dimensional materials such as SnS_2,_ with an energy gap in the visible light spectrum, and InSe, with high electron mobility, the combination of them can create a novel platform for electronic and optical devices. Herein, we study a tunable gain SnS_2_/InSe Van der Waals heterostructure photodetector. SnS_2_ crystals were synthesized by chemical vapor transport method and characterized using X-ray diffraction and Raman spectroscopy. The exfoliated SnS_2_ and InSe layers were transferred on the substrate. This photodetector presents photoresponsivity from 14 mA/W up to 740 mA/W and detectivity from 2.2 × 10^8^ Jones up to 3.35 × 10^9^ Jones by gate modulation from 0 V to +70 V. Light absorption and the charge carrier generation mechanism were studied by the Silvaco TCAD software and the results were confirmed by our experimental observations. The rather high responsivity and visible spectrum response makes the SnS_2_/InSe heterojunction a potential candidate for commercial visible image sensors.

## 1. Introduction

Image sensors are widely used in smartphones, webcams, security cameras, etc. [1,2]. Each pixel consists of a photodiode and an electronic circuit to amplify and transmit the signal. The fill factor, which is defined as the ratio of the light absorption to the total pixel area, is an important parameter in image sensor fabrication [3]. Improving the fill factor and controlling the photodiode gain for operation in the unsaturated region are important considerations [4]. In current technology, the gain is tuned according to the received light intensity by electronic components like transistors, capacitors, etc. However, these components occupy part of the pixel area, which in turn reduces the fill factor. To solve this challenge, various solutions have been proposed, including utilizing smaller transistors, which greatly increases the fabrication cost [4]. Another solution is replacing the photodiode with a photogate where carrier concentration (photocurrent) can be tuned by gate voltage modulation [5,6]. For efficient tuning, two-dimensional (2D) materials are introduced as an interesting candidate because their charge carriers are more affected by external electrostatic fields due to their atomic thickness [7]. For example, in most 2D materials-based transistors, a high switching ratio (~10^8^) and a suitable threshold slope (60 mV/dec) have been reported [8,9]. On the other hand, their high exciton binding energy and diverse energy gap provide the possibility of fabricating efficient photodetectors at different wavelengths [10]. Therefore, for the fabrication of tunable gain photodetectors, 2D materials homo or heterostructures configuration are suitable candidates [11].

There are some reports on tunable gain photodetectors based on Graphene, MoS_2_, ReS_2_, etc. For example, for graphene phototransistors, the observed gain decreases from 95 to 65 by applying a positive gate voltage, whereas it increases to 115 by applying a negative gate voltage. This behavior was attributed to the hole impact ionization effect by the external electrical field (gate voltage) [12]. In a graphene phototransistor fabricated on SiC substrate, photoresponsivity increases to a large value (7 A/W by negative gate voltage) due to high effective electric field caused by the substrate [13]. In another study, change in the effective barrier height in the graphene/InN heterostructure photodetector enhances the photoresponsivity five times by applying a positive gate voltage [14]. Due to having a variable energy gap, 2D semiconductor materials compared to graphene have more suitable optical and electronic properties. In the back gated n-type MoS_2_ (ReS_2_) based phototransistors, by applying positive gate voltages (accumulation area), the photoresponsivity increases two (three) orders of magnitude, whereas in negative gate voltage (depletion area) it is decreased by one order of magnitude [15].

Due to the weak Van der Waals bonding between 2D layers and the free dangling bonds, it is possible to fabricate ideal interfaces without lattice mismatch. The diversity of 2D materials creates distinct electronic and optical properties when putting these layers together, which promise the fabrication of efficient devices such as photodetectors with high photoresponsivity and photodetectivity at different wavelengths.

In this work, an SnS_2_/InSe heterostructure photodetector has been fabricated. The 2D InSe layers indicate a 1.5 eV energy gap and high electron mobility (~10^4^ cm^2^/Vs) [16,17], whereas SnS_2_ presents a 2.44 eV energy gap suitable for visible light spectrum absorption [18]. In this study, an attempt has been made to combine the high mobility of InSe and the proper energy gap of these two materials to fabricate a gain tunable efficient photodetector. SnS_2_ crystals are synthesized by the Chemical Vapor Transport (CVT) technique [19] and they were mechanically exfoliated on a SiO_2_/Si substrate. To fabricate the photodetector, a mechanically exfoliated InSe layer was transferred on the SnS_2_ layer. Our results indicate that the photoresponsivity and photo-detectivity are increased up to 50 times by positive gate voltage. This work demonstrates that SnS_2_/InSe heterojunctions can serve as qualified candidates for the next generation of tunable gain visible spectrum image sensors.

## 2. Materials and Methods

SnS_2_ crystals were grown by the CVT technique as follows. Tin and Sulfur powder with stoichiometric ratio were placed in a clean vacuum sealed quartz ampoule in a furnace at 850 °C for 2 weeks. After removing the grown crystal, its quality was investigated using the X-ray diffraction (Spectro Xepos, Kleve, Germany) and Raman (Horiba XploRA, Kyoto, Japan) techniques. To prepare the 2D layer, the crystal was mechanically exfoliated and transferred on SiO_2_/Si substrates [20]. The InSe layer, which was already exfoliated from a commercial high-quality crystal (HQ Graphene Co., Groningen, Netherland) on PMMA/PVA film, was transferred on the mentioned SnS_2_ layer using a home-made transferring system [21]. Finally, Cr/Au electrodes were deposited on both sides of the SnS_2_/InSe heterostructure by standard photolithography process and an electron gun evaporator.

Electrical characterization was measured using a Keithley 6487 picometer instrument. To investigate the optical properties of devices in the visible spectrum, we used three blue (460 nm), green (525 nm), and red (625 nm) LEDs with an optical power of 1 µW.

## 3. Results and Discussion

Figure 1a and b show the schematic and the optical microscope image of the SnS_2_/InSe photodetector, respectively, which has a channel length (L) of about 20 μm. Figure 1c shows the X-ray diffraction spectrum of the SnS_2_ crystals grown by the CVT technique. Comparing this pattern with standard tables [22] indicates suitable crystal quality. Figure 1d demonstrates the Raman spectrum including the A_1g_ characteristic peak of SnS_2_ at 314 cm^−1^ [23].

The energy band diagram of the device is shown in Figure 2a. Due to small differences in their work functions, electron transfer from InSe to SnS_2_ results in a small potential barrier (around 40 meV) and linear current versus voltage behavior (Figure 2b). Figure 2c shows the device drain current versus gate voltage. It behaves like an n-type transistor, as we expect based on type of carriers in both layers. The on/off current ratio is about 40, which indicates proper gate modulation in the field effect transistor. The equation μ = [dI_sd_/dV_g_] × [L/(W × C × V_sd_)] [24] has been used to calculate the field-effect mobility of the charge carriers. The maximum mobility was about 6 cm^2^/Vs for our device with length, L = 20 μm and width, W = 30 μm for device channel, and C as the capacitance/m^2^ of the gate oxide about C = 1.15 × 10^−4^ F/m^2^ for the silicon substrate with 300 nm SiO_2_ dielectric layer. Here, V_g_, V_sd_, and I_sd_ represent the applied back gate voltage, source-drain voltage, and current respectively. According to Figure 2c (inset), by applying the positive gate voltage (V_g_ > V_th_) the energy band of the layers shift downward, leading to accumulation of electrons within the conduction band of SnS_2_. Thus, applying a drain voltage results in a high device current, as is shown in the right part of Figure 2c.

The photocurrent (I_ph_ = I_light_ − I_dark_) of the device under illumination by three light sources with wavelengths of 460, 525, and 625 nm at the optical power of 1 µW is shown in Figure 3a. As is shown, shorter wavelengths, or more energetic photons for a given bias voltage, result in higher photocurrent. The depletion charge density can be calculated by Q_D_ = C × ∆V_th_/e, where C and ΔV_th_ are capacitance and threshold voltage shift, respectively. By using the carrier density, the photogain can be calculated through the G = I_ph_/(e × Q_D_ × S) [25]. According to Figure 3b, the threshold voltage shift is equal to 10 V for 460 and 525 nm wavelength and 5 V for 625 nm wavelength. By applying +70 V gate voltage, the calculated Q_D_ are ≈ 7.31 × 10^11^ cm^−2^ for both 460 and 525 nm and 3.65 × 10^11^ cm^−2^ for 625 nm and photogain are 1.05 × 10^10^, 8 × 10^9^, and 7.45 × 10^9^ for 460, 525, and 625 nm, respectively.

The photoresponsivity (R ≡ I_ph_/P_optical_) and detectivity of the devices is presented in Figure 4. As seen in Figure 4a, photoresponsivity at +1 V reaches to 14, 10, and 2.5 mA/W using the three light sources. Detectivity is defined as D* ≡ RS^1/2^/(2eI_dark_)^1/2^ where R is responsivity, S is effective area (here: ~ 600 μm^2^), and e is elementary charge. Detectivity reaches to 2.2 × 10^8^, 1.6 × 10^8^, and 5 × 10^7^ Junes at 460, 525, and 625 nm at under +1 V respectively. Increasing the charge density and photogain result in the photocurrent enhancement and thus improves the photoresponsivity up to 740 mA/W, 560 mA/W, and 260 mA/W and detectivity up to 3.35 × 10^9^ Junes, 2.55 × 10^9^ Junes, and 1.18 × 10^9^ Junes at V_g_ = +70 V for 460, 525, and 625 nm, respectively (Figure 4a). As shown in Figure 4b, by applying the gate voltage, the photoresponsivity rises. Therefore, we can control photodetector performance using the gate voltage. By applying a pulsed gate voltage (Figure 4c) without external electronic circuits, the unsaturated state image sensor with real-time tunable gain can be fabricated.

As can be seen, by applying a gate voltage, the photoresponsivity increases up to 50 times, which shows good sensitivity to the gate voltage and shows an improvement compared to similar structures. Table 1 shows a comparison between our devise and the previous reports [26,27,28,29].

To understand the process of light absorption and the mechanism of electron-hole pair generation, this structure was simulated by the Silvaco TCAD software. Figure 5a and b show the simulated structure and the carrier concentration at V_g_ = −5 and 5 V, respectively. It is observed that at V_g_ = +5 V the carrier concentration is higher than at V_g_= −5 V, which can increase the photocurrent at positive gate voltage. The carrier concentration is calculated through n = σ/eμ where σ is conductivity at V_g_ [30] and the photocurrent can be calculated by I_pc_ = eμnEWD, where E and D are the electric field in the channel and absorption depth, respectively [31]. At positive gate voltage, with increasing conduction (in large slope region in Figure 2c), carrier concentration is also increased. This results in a photocurrent enhancement, which had already been observed in experimental results.

The simulated photogeneration at the three wavelengths that were used is shown in Figure 6a. As can be seen, most of the photogeneration is in the SnS_2_ layer, due to more carriers of n-type SnS_2_ than InSe. The SnS_2_ layer has little absorption in the red light range because of rather high energy gap. As a result, in our device, the photocurrent will generate only by absorption of red light in the InSe layer, which causes a photoresponsivity of about 2.5 mA/W. However, in the blue and green light regions, both layers are active, which causes a significant increase in photocurrent and photoresponsivity (14 and 10 A/W for blue and green light). In this way, the SnS_2_/InSe heterostructure causes the broadband photodetector and works properly in the entire range of the visible light spectrum (Figure 6b).

## 4. Conclusions

In conclusion, an SnS_2_/InSe heterostructure photodetector for the visible light spectrum has been introduced. This photodetector presents photoresponsivity from 14 mA/W up to 740 mA/W and detectivity from 2.2 × 10^8^ Jones up to 3.35 × 10^9^ Jones by applying various gate voltages. According to experimental and simulation results, at 460 and 532 nm wavelengths, due to the proximity of the energy of the photon to the SnS_2_ energy gap, light is absorbed in both layers and the photocurrent increases. Our results showed that the gate voltage increases the carriers in the SnS_2_ layer. Due to the weak absorption of red light in the SnS_2_ layer, applying gate voltage does not affect the photocurrent at red light compared to blue and green light. The results show that this structure has potential for use in visible light tunable gain image sensors.

## Figures and Tables

**Figure 1 micromachines-13-02068-f001:**
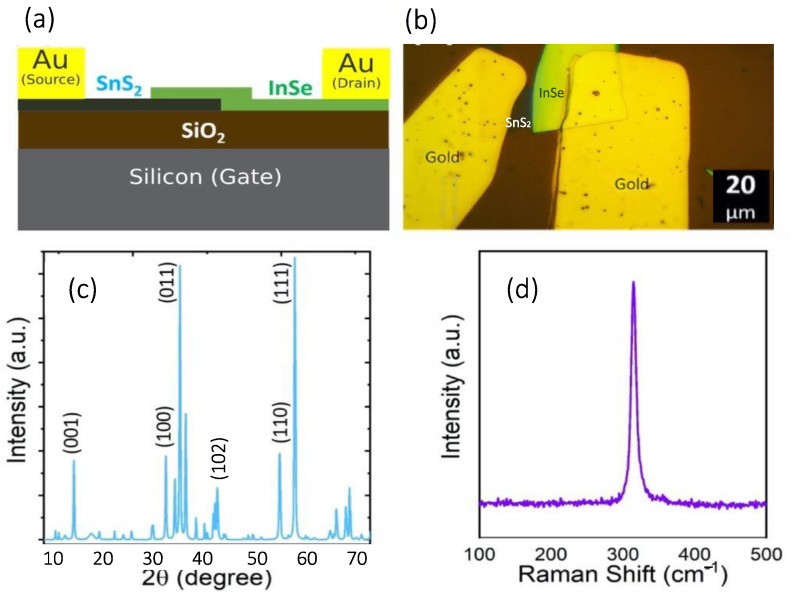
(**a**) Schematic of SnS_2_/InSe photodetector. (**b**) Optical microscope image of the fabricated device. (**c**) XRD spectrum of the SnS_2_ Crystal. (**d**) Raman Spectrum of the SnS_2_ layer.

**Figure 2 micromachines-13-02068-f002:**
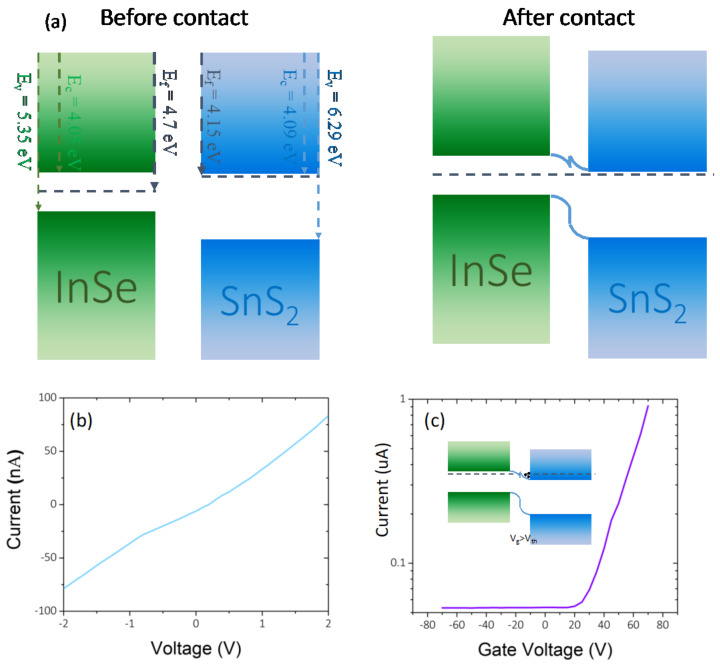
(**a**) Band diagram of SnS_2_/InSe heterostructure before and after contact, after contact, electron transfer from InSe to SnS_2_ and hole transfer from SnS_2_ to InSe (E_f_, E_c_, and E_v_ are Fermi level, conduction band edge, and vallance band edge, respectively, The energy values are shown in figure). (**b**) Typical current-voltage curve of the device. (**c**) Drain current versus gate voltage of the device. Inset presents band diagram in positive gate voltage.

**Figure 3 micromachines-13-02068-f003:**
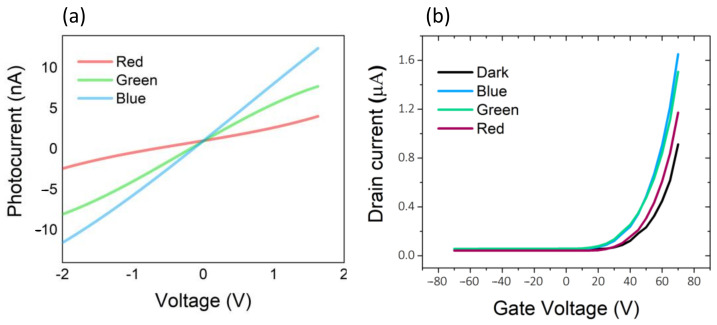
(**a**) Photocurrent of the device under light irradiation. (**b**) The transfer curves in light irradiations.

**Figure 4 micromachines-13-02068-f004:**
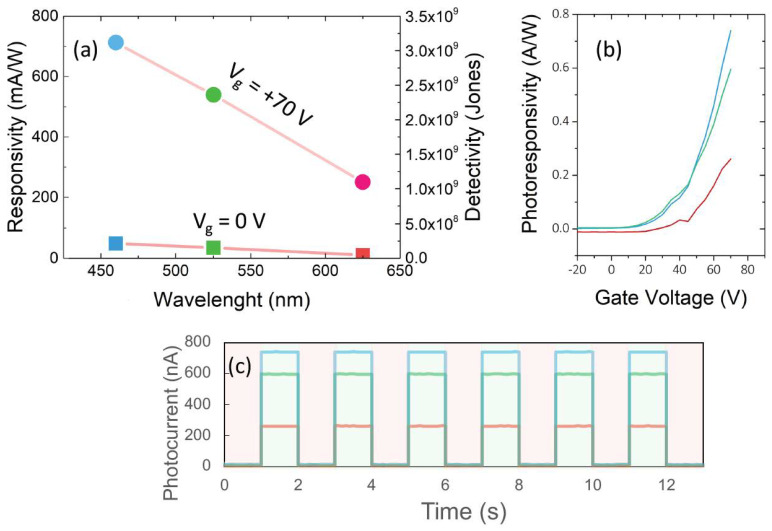
(**a**) photoresponsivity and photodetectivity of the device under light irradiation by applying and without applying gate voltage; (**b**) photoresponsivity versus gate voltage under red, green and blue light irradiation (**c**) photocurrent of the device under pulsed applying gate voltages at different wavelength irradiation (green is applying +70 V gate voltage and red is without gate voltage).

**Figure 5 micromachines-13-02068-f005:**
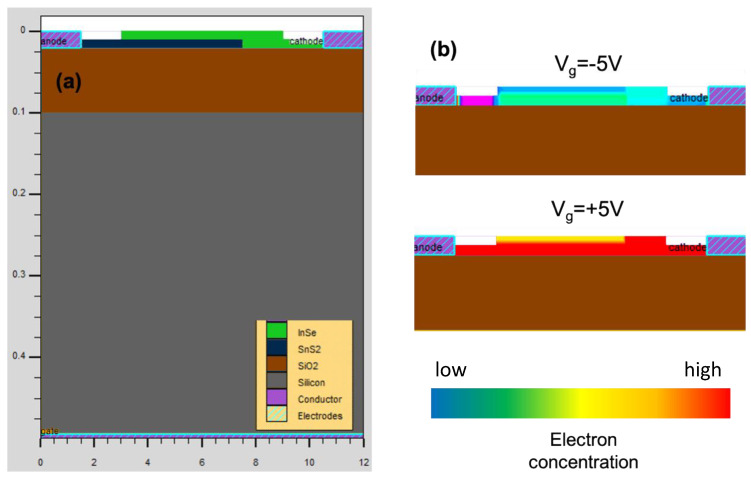
(**a**) Schematic of the simulated structure (500 nm × 12 μm). (**b**) Carrier concentration at positive and negative gate voltage in the channel.

**Figure 6 micromachines-13-02068-f006:**
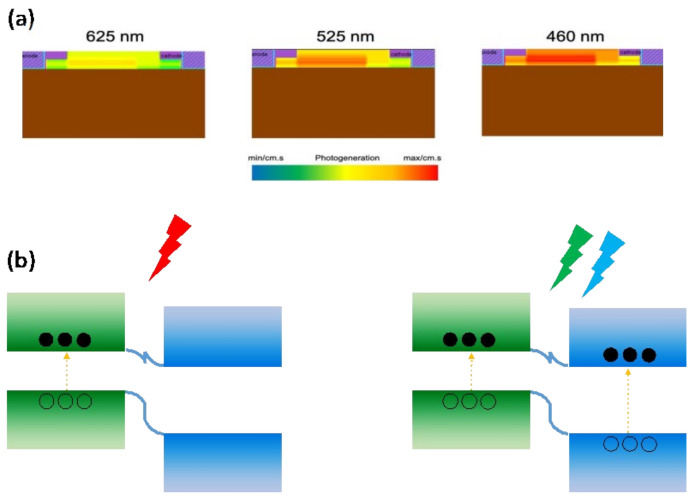
(**a**) Simulated photogeneratin at 625 nm, 525 nm and 460 nm wavelength. (**b**) Band diagram of device in light irradiation.

**Table 1 micromachines-13-02068-t001:** Comparison of gate effect on 2D photodetectors.

Structure	Gain	Photoresponsivity Ratio (Applying Gate/without Applying Gate)	Response Wavelength (nm)	References
Graphene	115	1.16	632	[12]
Graphene/InN	-	4	550	[14]
Graphene/MoS_2_	10^8^	5	650	[26]
InSe	-	2.5	254~850	[27]
SnS_2_	-	3	300~750	[28]
SnS_2_/InSe	-	1.4	405	[29]
SnS_2_/InSe	10^10^	52	460~625	This Work

## Data Availability

Not applicable.

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
