# Peer review of "Tunable Gain SnS2/InSe Van der Waals Heterostructure Photodetector"

_micromachines, 2022, doi:10.3390/mi13122068_

Round 1
Reviewer 1 Report
I have reviewed the article entitled “Gain tunable SnS2/InSe Van der Waals heterostructure photo-detector”. The authors have reported on SnSe/InSe heterostructure photo-detector.
Comments to author;
1. Line 15 “0V to +70V” should be 0 V and +70 V
2. Line 30-32 requires grammatical improvement “including utilize of technology nodes with smaller transistors, which greatly increases the fabrication cost”
3. Similarly, line 31-32 requires grammatical improvement “Another solution is replacing the photodiode with a photogate that carrier concentration (photocurrent) can be controlled by gate voltage modulation”
4. Line 33, “For efficient tuning, materials and structures are most important” This sentence has no meaning and connection to the previous one. Make it meaningful or delete.
5. Line 34-34, “Two-dimensional (2D) materials are introduced as an interesting candidate because their charge carriers are more affected by external electrostatic fields due to their atomic thickness.” The author should specify the 2D materials, substantiate your statements with examples.
6. Line 50 “photoresponsivity five times by applying gate voltage” Specify if it is positive or negative gate voltage.
7. Line 62, “cm2/Vs” 2 should be superscript
8. Line 69, “30 times by gate voltage” Specify whether it is positive or negative gate voltage.
9. Line 103 “6 cm2/V.s.” the two dots should be removed.
10. Figure 2a, the author should indicate conduction band, valence band, electron affinity …energy levels of SnSe and InSe. They can get these values from reliable literature sources and draw a reliable energy band diagram.
11. Line 122 “10V, 10V and 0V” should be written as 10 V, 10 V and 0 V. It is also not clear if 10 V is repeated twice. Please clarify.
12. Line 123 “+70V gate voltage,” should be written as +70 V. Check the whole document. Similar errors exist and should be corrected.
13. Line 123 “QD are ≈7.31 × 1011” 11 could be a superscript. Cross check.
14. Line 134 “+70V gate voltage” should be written as +70 V.
15. Line 143 “where E and D are electric field in the channel and absorption 143 depth respectively” Define W
16. Re-scale figure 1c. Use a scale of 10 on the horizontal axis. The authors should index or identify the un-indexed peaks.
17. The authors should provide the thickness of SnS2, InS and Cr/Au electrodes. Include this information at appropriate places.
18. The authors should extract the response/recovery time of their photo-detector from Figure 3f and compare with others in the literature.
19. Discuss your results with related hetero-structures in the literature. For comparison, you can make a present the important parameters of your photodetector in tabular form.

Author Response
Dear Reviewer,
Thanks for your comments, the response to your comments has been uploaded to the Word document.

Reviewer 2 Report
In this work, the authors report gain tunable SnS2/InSe Van der Waals heterostructure photodetector. However, more details of this manuscript should be improved. There some problems need to be work out before the publication of the article. The following are several major comments and questions for the authors:
1. In this paper, the carrier concentration is regulated by gate voltage and the current is increased to illustrate the generated gain. However, under gate voltage, the light and dark current will increase simultaneously. The author should provide more data to prove the generated gain. And author should figure out what the gain is.
2. The band diagram will change when gain occurs, and the author should add relevant content.
3. In Figure 2(c), The transfer curve suggestion is expressed in log coordinates, which will be more intuitive.
4. In this work, the transfer curve of separate SnS2, InSe and SnS2/InSe heterojunction should be shown, so as to further explore the internal physical mechanism. Please refer to the article (ACS Nano 2022, 16, 10, 17347-17355 and Advanced Materials, 2022, 34, 2203283).
5. It is found that the absorption of red light is relatively weak in the experiment and simulation. The simulation reason mentioned is that only SnS2 is involved in absorption, so why should we prepare this heterojunction? Is it because the gate voltage cannot regulate a single SnS2 photodetector? Obviously not, which has been reported in relevant literature (Advanced Functional Materials, 2020, 30(24): 2001650.)
Author Response

(The authors gave the same response as above.)

Round 2
Reviewer 1 Report
None